# Reprogrammed CD8^+^ T-Lymphocytes Isolated from Bone Marrow Have Anticancer Potential in Lung Cancer

**DOI:** 10.3390/biomedicines10061450

**Published:** 2022-06-19

**Authors:** Evgenii G. Skurikhin, Olga Pershina, Natalia Ermakova, Angelina Pakhomova, Darius Widera, Mariia Zhukova, Edgar Pan, Lubov Sandrikina, Lena Kogai, Nikolai Kushlinskii, Sergey G. Morozov, Aslan Kubatiev, Alexander Dygai

**Affiliations:** 1Laboratory of Regenerative Pharmacology, Goldberg ED Research Institute of Pharmacology and Regenerative Medicine, Tomsk National Research Medical Centre of the Russian Academy of Sciences, Lenin, 3, 634028 Tomsk, Russia; ovpershina@gmail.com (O.P.); nejela@mail.ru (N.E.); angelinapakhomova2011@gmail.com (A.P.); mashazyk@gmail.com (M.Z.); artifexpan@gmail.com (E.P.); ermolaeva_la@mail.ru (L.S.); kogay-lena@mail.ru (L.K.); amdygay@gmail.com (A.D.); 2Stem Cell Biology and Regenerative Medicine Group, School of Pharmacy, Whiteknights Campus, Reading RG6 6AP, UK; d.widera@reading.ac.uk; 3Blokhin National Medical Research Center of Oncology, 115522 Moscow, Russia; kne3108@gmail.com; 4Institute of General Pathology and Pathophysiology, 125315 Moscow, Russia; smorozov.biopharm@mail.ru (S.G.M.); akubatiev.niiop@mail.ru (A.K.)

**Keywords:** Lewis lung carcinoma, reprogramming, MEK inhibitors, cancer immunotherapy

## Abstract

CD8^+^ T-lymphocytes play a key role in antitumor immune response. Patients with lung cancer often suffer from T-lymphocyte dysfunction and low T-cell counts. The exhaustion of effector T-lymphocytes largely limits the effectiveness of therapy. In this study, reprogrammed T-lymphocytes used MEK inhibitors and PD-1 blockers to increase their antitumor activity. Antitumor effects of reprogrammed T-lymphocytes were shown in vitro and in vivo in the Lewis lung carcinoma model. The population of T- lymphocytes with persistent expression of CCR7 was formed as a result of reprogramming. Reprogrammed T-lymphocytes were resistant to apoptosis and characterized by high cytotoxicity against Lewis lung carcinoma (LLC) cells in vitro. Administration of reprogrammed T-lymphocytes to C57BL/6 mice with LLC reduced the number of lung metastases. The antitumor effect resulted from the elimination of tumor cells and cancer stem cells, and the effect of therapy on cytotoxic T-lymphocyte counts. Thus, reprogramming of T-lymphocytes using MEK inhibitors is a promising approach for targeted therapy of lung cancer.

## 1. Introduction

Lung cancer is the leading cause of cancer-related deaths among men and women [1]. Major advances have been made using targeted therapies and checkpoint inhibitors, which improve the survival of patients with lung cancer [2]. However, some forms of lung cancer remain resistant to treatment.

Cytotoxic CD8^+^ T-lymphocytes are the most powerful effector cells in immune response, which play a key role in antitumor immunity. CD8^+^ T-lymphocytes induce tumor cell death by producing cytotoxic molecules such as perforin and granzymes [3]. In addition, interferon-γ secreted by CD8^+^ T-lymphocytes enhance antitumor properties of other immune cells [4]. However, T-lymphocytes become dysfunctional in the tumor microenvironment. In addition, CD8^+^ T-lymphocyte counts are significantly lower in patients with lung cancer compared with healthy individuals. High levels of CD8^+^ T cells after lung cancer therapy may be a predictor of better survival rates in patients [5].

Tumor and tumor microenvironments suppress proliferation of CD8^+^ T-lymphocytes and decrease their numbers [6]. Interestingly, during interaction with the tumor, CD8^+^ T-lymphocytes have a decreased activity of metabolic pathways and hyporeactive phenotypes that cannot be restored by stimulation [7]. These changes limit the use of effector T-lymphocytes in immunotherapy of lung cancer.

Forced changes in the intracellular metabolism of immune cells is a promising approach to increase the effectiveness of antitumor response. Importantly, in contrast to genetic reprogramming, metabolic reprogramming of cells does not lead to changes in cellular identity [8]. Guo et al. showed that IL-10 influences the restoration of cytotoxic function and proliferative activity of exhausted CD8^+^ T-lymphocytes [9]. In another study, Verma et al. demonstrated that inhibition of the MAPK/ERK signaling pathway causes the metabolic reprogramming of effector CD8^+^ T-lymphocytes and enhances their antitumor activity [10].

The mechanism of action of MEK1/2 inhibitors is based on the altering of T-lymphocyte metabolism by modulating the MAPK/ERK1/2 metabolic pathway. Activation of T-lymphocyte receptors prepares cells for division. However, transient inhibition of MEK1/2 reduces levels of activated ERK1/2 and cyclin D1. This arrests the cell cycle and enhances PGC1α/SIRT3, followed by enhanced metabolism by increasing fatty-acid oxidation. At the same time, low levels of activated ERK1/2 and cyclin D1 do not affect antigen-mediated T-lymphocyte activation. As a result, a population of memory T-lymphocytes similar to stem cells (Tscm) is generated. Tscm are cells that are phylogenetically positioned between naive T-lymphocytes and memory T-cells [7]. These cells differ from central memory cells (Tcm) by higher methylation of effector genes, self-renewal, multipotency, ability to proliferate, and an overexpression of CCR7 and CD62L. MEKi-induced Tscm cells show strong cellular activation, high antigen-specific responses, and long-term survival. It should be noted that effector memory CD8^+^ T-lymphocytes (T_EMs_) lack CCR7 and CD62L [11].

The restoration of the cytotoxic function of T-lymphocytes is also associated with the PD-1/PD-L1 signaling pathway. Under physiological circumstances, the PD-1/PD-L1 signaling pathway maintains immunological tolerance and prevents the development of autoimmune processes. In cancer, PD-1 is involved in the inhibition of antitumor immunity by stimulating apoptosis and reducing synthesis of perforins, IFN-γ, IL-2, and TNF-α [12]. PD-1 activation alters T-lymphocyte metabolism by impairing glucose utilization and by inhibiting amino-acid metabolism [13]. In contrast, inhibition of PD-1 has been shown to restore the effector function of T-lymphocytes [14].

Actively proliferating cancer cells are the main targets of anticancer treatments. This leads to a partial elimination of the tumor. However, cancer stem cells (CSC) with low metabolism can escape the therapy and survive. CSCs are responsible for tumor development, heterogeneity, metastasis, and recurrence. Therefore, T-lymphocyte therapy should be focused on elimination of CSCs, as well as tumor cells. Thus, CSCs are potential therapeutic targets and diagnostic and prognostic markers for lung cancer [15]. This approach may be more effective in the early stage of tumor development. In lung cancers, different CSC populations have been identified according to the expression of some selected markers, such as CD44, CD117, EGF, Axl, and Sox2 [16]. An attempt has already been made to study the interaction of CSCs and CD8^+^ T cells [5]. We suggest that CSCs and tumor cells are potential targets for reprogrammed T-lymphocytes. In the present study, we assessed the therapeutic potential of inhibiting the MAPK/ERK pathway with MEK1/2i to reprogram CD8^+^ T-lymphocytes for the use in Lewis lung carcinoma cell therapy in vitro and in vivo. The PD/PDL-1 signaling pathway was blocked by the immune checkpoint inhibitor in order to protect T-lymphocytes from the immunosuppressive effect of LLC.

## 2. Materials and Methods

### 2.1. Animals

Male C57BL/6 mice that were 8–10 weeks old were obtained from the nursery of the Experimental Biological Models Department of the E. D. Goldberg Research Institute of Pharmacology and Regenerative Medicine (veterinary certificate is available). Animal keeping and design of experiments were approved by the Ethics Committee of the E. D. Goldberg Research Institute of Pharmacology and Regenerative Medicine (protocol No. 189092021). The animals were maintained in accordance with the European Convention for the Protection of Vertebrates (Strasbourg, 1986); Principles on Good Laboratory Practice (OECD, ENV/ MC/ CUEM (98)17, 1997).

### 2.2. Lewis Lung Carcinoma Cell Line 

The Lewis lung carcinoma (LLC) cell line of C57BL strain was used in experiments in vivo and in vitro (400263 CLS Cell Lines. Service, GmbH, Köln, Germany). LLC cells were established from the spontaneous lung adenocarcinomas that occur in C57BL/6 mice.

### 2.3. Lewis Lung Carcinoma Cell Culture

The LLC cells were plated at a seeding density of 3 × 10^5^ cells/1 cm^2^ in the T-25 flask. The cells were maintained in RPMI 1640 medium supplemented with 2 mM L-glutamine and 10% fetal bovine serum (FBS, Sigma-Aldrich, St. Louis, MO, USA) at 37 °C in a humidified atmosphere containing 5% CO_2_. The culture medium was changed 2–3 times per week.

### 2.4. Orthotopic Model of Lewis Lung Carcinoma

Under sterile conditions, mice fur was removed on the left side of the chest above the lower line of the ribs and just below the lower border of the scapula. Each animal was injected with 50 µL suspension of LLC cells (1.5 × 10^6^ cells) by an insulin syringe with 30 G needle to a depth of 5 mm into the left lung between the 6th and 7th rib [17,18]. All manipulations were accomplished with isoflurane inhalation using an inhalation anesthesia machine (UGO BASIELE, model 21050, Comerio, Italy). After 7 days of LLC cell inoculation, the animals were euthanized.

### 2.5. Study Design

The study design is shown in Figure 1. In the first stage, we carried out reprogramming of CD8^+^ T-lymphocytes derived from the bone marrow. Next, we evaluated migration of reprogrammed CFSE-labeled CD8^+^ T-lymphocytes into the lungs of the recipient mice. The cells were injected into the tail vein. In the third stage, we assessed cytotoxic activity of reprogrammed CD8^+^ T-lymphocytes in the LLC cell culture in vitro. Additionally, we studied apoptosis of reprogrammed CD8^+^ T-lymphocytes. Finally, the antitumor and antimetastatic activity of reprogrammed CD8^+^ T-lymphocytes was evaluated in vivo.

### 2.6. Isolation of Mononuclear Cells

Mononuclear cells from blood, lungs, and bone marrow were isolated as described earlier [19,20].

### 2.7. Flow Cytometry

Expression of membrane and intracellular receptors of cancer stem cells and lymphocytes derived from the blood and the lungs was studied using mouse monoclonal antibodies following standard flow-cytometry protocols. Briefly, cell suspension was pre-incubated for 5 min with antimouse CD16/CD32 (BD FcBlock™). After pre-incubation, the cell suspension was stained with fluorophore-conjugated monoclonal antibodies: CD3 PerCP, CD8 BV510, CD44 APC-Cy™7, CD45 PerCP, CD45RA PerCP-Cy™5.5, CD69L APC, CD117 FITC, CD95 BV421, EGF (F4/80) Alexa Fluor^®^ 647, Axl BV421, CD279 (PD-1) BV421, CD197 (CCR7) PE (all—1/50 dilution, BD Biosciences, San Jose, CA, USA). The relevant isotype controls were used. Further, the cell suspension was stained with Sox2 PE intracellular antibodies (1/50 dilution, BD Biosciences, San Jose, CA, USA). FACS Canto II flow cytometer with FACS Diva software was used for analysis (BD Biosciences, Franklin Lakes, NJ, USA). 

### 2.8. Magnetic Separation of CD8^+^ T-Lymphocytes

After isolation of mononuclear cells from the bone marrow, a magnetic separation was performed to enrich the cell fraction with CD8^+^ T-lymphocytes. The cell suspension was enriched with naive CD8^+^ T-lymphocytes (CD3^+^CD8^+^CD44^−^CD62L^+^) by magnetic separation. Enrichment was performed following a standard protocol using a mouse kit (EasySep^TM^ Mouse Naive CD8^+^ T Cell Isolation Kit, as recommended by the manufacturer (StemCell Technologies, Vancouver, BC, Canada).

### 2.9. Reprogramming of CD8^+^ T-Lymphocytes

After enrichment of the cell suspension with CD8^+^ T-lymphocytes using magnetic separation, the cells were incubated in the medium recommended for CD8^+^ T-lymphocytes (RPMI 1640 (Sigma-Aldrich, St. Louis, MO, USA) with the addition of 10% FBS (Sigma-Aldrich, USA), 2 mM L-glutamine (Sigma-Aldrich, St. Louis, MO, USA), 10 mM HEPES (Sigma-Aldrich, St. Louis, MO, USA), and 55 μM β-mercaptoethanol (Thermo Scientific™ 35602BID, Thermo Scientific, Waltham, MA USA), 37 °C, 5% CO_2_) for 2–3 h (Appendix A). The concentration of T-lymphocytes was 1 × 10^8^/mL. The volume of the medium in the vial was at least 5 mL.

An antigen-presenting mix was prepared from LLC cell lysate by using a freeze–thaw cycle in 0.85% NaCl solution. The cycle was repeated five times in rapid succession from −70 °C to 37 °C, and then refrozen and stored at −70 °C before use (Figure 1). After the final thawing, the lysate was stained by trypan blue (Sigma-Aldrich, St. Louis, MO, USA) [21]. The preparation of adjuvant (Freund’s adjuvant) for the antigen-presenting mix was carried out according to the manufacturer’s standard protocol (Sigma-Aldrich, St. Louis, MO, USA). Freund’s adjuvant solution was mixed with the tumor cell lysate (3 × 10^4^/mL) at a 1:1 ratio to form a thick emulsion (Appendix A).

CD8^+^ T-lymphocytes were pre-incubated for 2 h before reprogramming. For reprogramming, 50 µL of an antigen-presenting mix with 1 µM MEK inhibitor was added to a flask with CD8^+^ T-lymphocytes of a given population (the concentration of CD8^+^ T-lymphocytes was 1 × 10^8^/mL, the volume of the medium in the flask was at least 5 mL). The resulting cell suspension was incubated for 48 h at 37 °C and 5% CO_2_. Reprogrammed CD8^+^ T-lymphocytes of a given phenotype were incubated for 2 h with human monoclonal antibody (MAT) nivolumab at a concentration 10 µg/mL in order to protect cells from the humoral action of LLC. At the end of the incubation cycle, suspensions were washed 2 times in the medium recommended for CD8^+^ T-lymphocytes (Appendix A). Immunophenotype and cytotoxicity of reprogrammed CD8^+^ T-lymphocytes were analyzed (Cytation 5).

To estimate the population stability, an exhaustion of reprogrammed CD8^+^ T-lymphocytes was carried out in vitro [22]. Reprogrammed CD8^+^ T-lymphocytes (1–3 × 10^6^ cells/mL in the medium supplemented with IL-2 (30 IU/mL) (StemCell Technologies, Vancouver, BC, Canada) were stimulated by T-Activator CD3/CD28 Dynabeads^®^ (Thermo Fisher Scientific Baltics, Vilnius, Lithuania) at a particle-to-cell ratio of 1:1. Restimulation was performed every 48 h 3–4 times (Appendix A). Then, the Dynabeads were removed. Immunophenotype and cytotoxicity of reprogrammed CD8^+^ T-lymphocytes were analyzed.

### 2.10. CFSE Staining of CD8^+^ T-Lymphocytes

CFSE labeling of CD8^+^ T-lymphocytes isolated from bone marrow of donor mice was used to determine the content of CD8^+^ T-lymphocytes in the lungs of recipient mice. CFSE staining of CD8^+^ T-lymphocytes was performed according to the manufacturer’s instruction (BD Biosciences, San Jose, CA, USA). All experimental samples, including controls, were analyzed using the same instrument settings. The cell population of intact control animals served as a negative control. The percentage of CFSE-positive cells in the resulting population was determined by analyzing data in flow-cytometry software, excluding doublets and other cell conglomerates from the analysis.

### 2.11. CD8^+^ T-Lymphocytes Injection

To evaluate the migration of CD8^+^ T-lymphocytes into the lungs of recipient mice from intact control and recipient mice with LLC, CFSE-labeled reprogrammed and nonreprogrammed CD8^+^ T-lymphocytes were injected intravenously at 1 × 10^6^ cells/mouse in 0.1 mL PBS once.

To assess the antitumor and antimetastatic activity, CD8^+^ T-lymphocytes were administered intravenously to recipient mice with LLC at 1 × 10^6^ cells/mouse in 0.1 mL of PBS on the 4th and 6th days of the experiment.

### 2.12. Detection of the CCR7 Expression, Cytotoxicity, and Apoptosis of CD8^+^ T-Lymphocytes In Vitro

Images of cells were obtained using the cell-imaging Cytation 5 (BioTek Instruments, Inc., Winooski, VT, USA) equipped with the following cubes: DAPI (blue), GFP (green), RFP (yellow). 

To assess CCR7 expression, T-lymphocytes were stained with anti-CCR7 antibodies and polyclonal secondary antibody donkey anti-Rabbit IgG H&L Alexa Fluor^®^ 555 (all Abcam, Cambridge, MA, USA). Nuclei were additionally stained with Hoechst 34580 (blue); CD8 FITC (green) was used to CD8^+^ T-lymphocyte detection. The percentage of CD8+CCR7+ cells were determined as the ratio of cells counted in green and red channel to total cells counted using blue (DAPI) channel.

Cytotoxicity and apoptosis of CD8^+^ T-lymphocytes were studied in cell culture of LLC. After co-incubation, CD8^+^ T-lymphocytes and LLC cells were stained with Hoechst 34580 (for nuclear staining) and 7-AAD (for apoptotic cells detection). Preliminary CD8^+^ T-lymphocytes were stained with CFSE BD Horizon. Cytotoxicity of CD8^+^ T-lymphocytes in LLC culture was assessed by analyzing the ratio of cells counted in the blue and red channels to the total number of LLC (percentage of dead Hoechst^+^7AAD^+^ LLC). Determination of the percentage of dead T-lymphocytes Hoechst^+^CFSE^+^ was made by the ratio of cells counted in blue and green channel to total cells.

All images were obtained with Cytation 5 (4× or 20× magnification) followed by cell analysis using Gen5™ data-analysis software (BioTek, Instruments, Friedrichshall, Germany). Prior to the analysis, images were preprocessed to align the background. 

### 2.13. Histological Examination of the Lungs

Lung preparations were fixed in 10% neutral buffered formalin, passed through increasing concentrations of alcohol to xylene and embedded in paraffin wax according to the standard method, then sectioned into 5 μm-thick slices, and stained with hematoxylin and eosin [20,23]. 

### 2.14. Assessment of Tumor Growth

The effect of cell therapy on LLC growth was assessed by statistical comparison of the tumor node volume in the control and experimental groups at the different observation periods, according to the duration of tumor growth retardation and tumor growth inhibition index (TGII) [24]
TGII = (Vc − V_O_)/Ve × 100%
where Vc and Ve are the average node volume in the control and experimental groups. 

### 2.15. Assessment of Tumor Volume

Linear dimensions of tumor nodes were measured in orthogonal planes and their volume was calculated in the elliptical approximation [25]. The tumor was measured with a caliper and the volume of the tumors was calculated by the formula: V = π/6 × length × width × height

### 2.16. Statistical Analysis

Statistical analysis was performed by methods of variational statistics using the SPSS 12.0 software (SPSS Inc., Chicago, IL, USA). The arithmetic mean (M), error of the mean (m), and the probability value (*p*) were calculated. The difference between the two compared values was significant at *p* < 0.05.

## 3. Results

### 3.1. Study of CCR7 Marker Expression by Reprogrammed CD8^+^ T-Lymphocytes of C57BL/6 Mice Bone Marrow In Vitro

We studied the expression of chemokine receptor CCR7 by CD8^+^ T-lymphocytes in vitro to evaluate the effectiveness of the reprogramming (Figure 2). Reprogrammed CD8^+^ T-lymphocytes have higher expression levels of chemokine receptor CCR7 than naive CD8^+^ T-lymphocytes in CD8^+^ T-lymphocyte culture (Figure 2). Exhaustion of reprogrammed CD8^+^ T-lymphocytes was performed to assess the persistence of the changes caused by the MEK inhibitor and nivolumab. Exhaustion did not cause changes in the CCR7 expression by reprogrammed CD8^+^ T-lymphocytes, which indicates that the changes induced by the MEK inhibitor and nivolumab are stable (Figure 2).

### 3.2. Study of Apoptosis of Naive and Reprogrammed CD8^+^ T-Lymphocytes in the Lewis Lung Carcinoma Cell Culture

#### 3.2.1. Control of CD8^+^ T-Lymphocytes

At the end of the cultivation cycle of naive CD8^+^ T-lymphocytes, the percentage of dead cells was 2.05% of the total cell number in the standard medium. Reprogrammed CD8^+^ T-lymphocytes are more stable under cultivation: 0.42% of the cells were in apoptosis (Figure 3).

#### 3.2.2. Cocultivation of CD8^+^ T-Lymphocytes with Cancer Cells

In a 0.25:1 dilution, we observed that naive CD8^+^ T-lymphocytes displayed increased apoptosis in LLC cell culture. The level of apoptosis was more than two-fold higher in comparison with that of the control lymphocytes.

An increase in the concentration of naive CD8^+^ T-lymphocytes in the LLC cell culture caused a decrease in apoptosis comparable with that in the control group.

Reprogrammed CD8^+^ T-lymphocytes of bone marrow were more resistant to the cytotoxic effect of cancer cells in comparison with naive CD8^+^ T-lymphocytes. This is indicated by a decrease in apoptosis of reprogrammed CD8^+^ T-lymphocytes in 0.25:1; 1:1, and 2.5:1 dilution (Figure 3).

These results confirm that CD8^+^ T-lymphocytes, when treated with MEKi and nivolumab in vitro, become T-cells, having high stable CCR7 expression and being resistant to the cytotoxic effect of cancer cells and distinct from naive T-lymphocytes.

### 3.3. Cytotoxicity of Reprogrammed CD8^+^ T-Lymphocytes Cocultured with Lewis Lung Carcinoma Cells 

The ratios of CD8^+^ T-lymphocytes to cancer cells, which were observed in the LLC cell culture, were 0:1, 0.25:1, 1:1, 2.5:1, 5:1, and 10:1. Naive and reprogrammed CD8^+^ T-lymphocytes induced apoptosis of cancer cells at the ratio of 0.25:1. The maximum level of tumor cell apoptosis was observed at a 10:1 cell ratio (CD8^+^ T-lymphocytes:cancer cells). Cytotoxicity of reprogrammed CD8^+^ T-lymphocytes was higher in comparison with that of naive CD8^+^ T-lymphocytes in the same ratios (Figure 4).

Together, these results demonstrate that MEKi not only results in increased CCR7 expression, but also enhances the functionality of reprogrammed CD8^+^ T-lymphocytes.

### 3.4. Evaluation of Migration of Naive and Reprogrammed CD8^+^ T-Lymphocytes Derived from Donor-Mice Bone Marrow into the Lungs of Recipient Mice with Lewis Lung Carcinoma

Naive and reprogrammed CD8^+^ T-lymphocytes derived from donor-mice bone marrow were preliminarily stained with vital dye CFSE. The analysis was performed 60 min after intravenous administration of CFSE-labeled CD8^+^ T-lymphocytes derived from donor-mice bone marrow to recipient mice with LLC.

Reprogrammed CD8^+^ T-lymphocytes of bone marrow migrated more actively to the lungs of mice with LLC. Thus, the number of CFSE-labeled reprogrammed CD45^+^CD8^+^ T-lymphocytes in the lungs of recipient mice with LLC was significantly higher than CFSE-labeled naive CD45^+^CD8^+^ T-lymphocytes (8.9 times) (Figure 5).

We found that reprogrammed with MEKi and nivolumab CD8^+^ T-lymphocytes were present at higher frequencies in the lung of mice with LLC after cell administration, compared to naive CD8^+^ T-lymphocytes.

### 3.5. Morphological and Histological Examination of the Lungs of Mice with Lewis Lung Carcinoma after Cell Therapy with Reprogrammed CD8^+^ T-Lymphocytes 

#### 3.5.1. Histological Examination

On the 7th day of the orthotopic model of LLC, well-vascularized large tumor nodes composed of atypical cells were detected in the lungs of C57BL/6 mice. This group of cells was characterized by cellular and nuclear polymorphism. Multinucleated giant cells and multiple cells in mitosis were found in the general population of atypical cells. There were numerous small foci of necrosis in the tumor tissue (Figure 6).

#### 3.5.2. Tumor Growth 

Therapy with reprogrammed CD8^+^ T-lymphocytes of bone marrow caused an increase in TGII. The value of TGII after cell therapy with reprogrammed CD8^+^ T-lymphocytes was 57%. Moreover, we observed an increase in tumor volume and the average number of metastases (Table 1). 

### 3.6. Effect of Reprogrammed CD8^+^ T-Lymphocytes on Cancer Cells and Cancer Stem Cells in the Lungs and Blood of Mice with Lewis Lung Carcinoma

The LLC model caused a significant increase in the number of CSCs with different phenotypes in the lungs of C57BL/6 mice on the 7th day of the experiment: Axl^+^, Axl^+^CD117^+^, EGF^+^CD44^+^Sox2^+^, EGF^+^Sox2^+^, CD44^+^Sox2^+^, CD117^+^Sox2^+^, CD117^+^EGF^+^CD44^+^Sox2^+^ (Figure 7). We observed an increase in the number of EGF^+^CD44^+^Sox2^+^, EGF^+^Sox2^+^, CD44^+^Sox2^+^, CD117^+^Sox2^+^, CD117^+^EGF^+^CD44^+^Sox2^+^ populations in the blood mice with LLC compared with the intact control on the d7, besides the Axl^+^ and Axl^+^CD117^+^ populations.

The injection of reprogrammed CD8^+^ T-lymphocytes derived from the bone marrow significantly reduced the number of cancer cells and CSCs population in the blood and lungs of recipient mice with LLC on the d7 (Figure 7). However, populations of cells Axl^+^ and Axl^+^CD117^+^ in the blood was increased after reprogrammed CD8^+^ T-lymphocyte administration. 

### 3.7. Effect of Reprogrammed CD8^+^ T-Lymphocytes on the Content of CD8^+^ T-Lymphocytes in the Blood and Lungs of Mice with Lewis Lung Carcinoma

We observed a decrease in the CD8^+^ T-lymphocyte population in the blood of C57BL/6 mice with LLC compared with intact control on the 7th day of the experiment: CD8^+^CD62L^+^CD197^+^CD45RA^+^, CD45RA^+^CD197^hi^CD62L^+^CD95^+^CD8^+^, CD8^+^CD62L^−^CD44^+^, CD3^+^CD8^+^, CD3^+^CD8^+^PD-1^+^ and CD3^+^CD8^+^PD-L1^+^ (Figure 8). At the same time, the number of CD8^+^ T-lymphocyte population with phenotype CD8^+^CD62L^+^CD197^+^CD45RA^+^, CD45RA^+^CD197^hi^CD62L^+^CD95^+^CD8^+^, CD3^+^CD8^+^CD3^+^CD8^+^PD-1^+^ andCD3^+^CD8^+^PD-L1^+^ increased in the lungs of mice with LLC. We explain this by increased migration of CD8^+^ T-lymphocytes from the blood to the lungs in response to tumor and exhaustion of CD8^+^ T-lymphocytes pool under tumor formation. The exception was effector-cytotoxic T-lymphocytes (CD8^+^CD62L^+^CD197^−^CD45RA^+^), whose number significantly decreased in the lungs of mice with LLC. We observed a declining trend in highly differentiated effector-memory T-lymphocytes (CD8^+^CD62L^−^CD44^+^) (Figure 8).

Injection of reprogrammed CD8^+^ T-lymphocytes caused an increase in the T-lymphocyte population in the blood of recipient mice with LLC compared to mice with LLC without treatment on the 7th day of the experiment (Figure 8). The exception was highly differentiated memory T-lymphocytes (CD8^+^CD62L^−^CD44^+^), whose number was lower compared to the intact control.

After T-lymphocyte therapy, the T-lymphocyte population in the lungs of recipient mice with LLC was lower in comparison with the animals of LLC group without treatment (Figure 8). At the same time, the number of CD45RA^+^CD197^hi^CD62L^+^CD95^+^CD8^+^, CD3^+^CD8^+^, CD3^+^CD8^+^PD-1^+^ and CD3^+^CD8^+^PD-L1^+^ cells were comparable to that in intact animals.

## 4. Discussion

Designing optimal (neo)adjuvant therapy is a crucial aspect of the treatment of lung cancer. Standard methods of chemotherapy, radiotherapy, and immunotherapy represent current standard strategies for treatment. However, in some cases with high metastatic activity and high levels of circulating tumor cells and CSC, the efficacy of standard treatment methods is insufficient and results in treatment failure, relapse, and ultimately reduced patient survival. Currently, much attention is paid to the search for targeted therapy of lung cancer. Targeted therapy for non-small-cell lung cancer (NSCLC) involves drugs that inhibit angiogenesis, and inhibitors of kinases EGFR, ROS1, MET, and RET. Targeted therapies for small-cell lung cancer (SCLC) are still limited compared with other forms of lung cancer. In some cases, PD-1/PDL-1 checkpoint inhibitors in combination with other therapies are used as targets for the treatment of SCLC [26]. Cancer vaccines and gene therapy for lung cancer are undergoing clinical trials [27]. Early cancer diagnosis, monitoring of the tumor growth progression and correct determination of histological subtype and tumor mutation are making treatment of lung cancer cumbersome and costly. Additionally, tumor heterogeneity, specific tumor microenvironment, and complex intercellular interaction between tumor and normal cells lead to the escape of the tumor from drugs and reduce the overall effectiveness of antitumor therapy.

Actively proliferating tumor cells are generally targets for drugs. Meanwhile, progression, metastasis, and recurrence of lung cancer, as well as resistance to therapy, are related to CSCs [15,28]. CSCs with a long resting state (G0 arrest) and high activity of the efflux transport system are insensitive to anticancer drugs [29]. Thus, treatment of lung cancer would largely benefit from early detection of CSCs and targeted CSCs elimination.

Currently, the search for therapies is aimed at enhancing antitumor immunity modifying immune cells. Development of CAR T-lymphocytes is one approach for the treatment of malignant tumors. CAR T-lymphocytes recognize specific tumor antigens and induce the eradication of tumor cells. CAR T-lymphocyte therapy has been approved for the treatment of blood cancers (leukemia and lymphoma). Clinical evaluation of efficacy and safety of TCR T-lymphocyte therapy in solid cancers (NCT04044859, NCT03132792, NCT03967223, NCT03907852), including NSCLC (NCT02592577, NCT02408016) is still ongoing [30]. The source of cells for modification is the patient’s lymphocytes. Persistent tumor antigen exposure generally causes an exhaustion of T-lymphocytes. Exhausted T-lymphocytes lose effector functions and persistently express inhibitory molecules such as PD-1, Tim-3, and CD160 [31]. Interestingly, modified CAR T-lymphocytes can be exhausted similarly to endogenous T-lymphocytes [32]. This prompts us to search for new approaches for cancer treatment.

MEK inhibition forms a pharmacologically feasible method that can be used to enhance different immune therapeutic approaches, including combination with priming agents, immune modulators or enhancing the efficacy of cellular therapies. Importantly, an MEK inhibitor is approved by the US Federal Drug Administration and is in use for anticancer treatment (NCT02407405, NCT00888134, NCT03109301) [10]. However, a lack of specificity and frequent side effects after systemic administration of a MEK inhibitor have been observed [7]. We propose an approach to reprogramming that involves combined use of MEK and PD-1 inhibitors. Bone marrow CD8^+^ T-lymphocytes were targets for modification. Inhibition of MEK1/2 leads to alteration of T-lymphocyte metabolism by modulating the ERK1/2 metabolic pathway. As a result, population of T memory stem cells (Tscm) has been generated [33]. MEKi-induced Tscm cells showed strong cellular activation, high antigen-specific responses, and long-term survival. Verma V et al. showed that MEK inhibition improved mitochondrial function of T-lymphocytes, inducing the formation of Tscm with CD45RA^+^CD197^hi^CD62L^+^CD95^+^CD8^+^ phenotype [10]. In order to protect T-lymphocytes from the immunosuppressive effect of tumors, we considered it appropriate to use an immune checkpoint inhibitor to block the PD/PDL-1 signaling pathway. Another requirement for successful reprogramming was the “training” of CD8^+^ T-lymphocytes with the entire spectrum of tumor antigens. The full set of antigens contained in the tumor cell lysate during training provided access for immune cells to a whole spectrum of epitopes for the formation of a tumor-specific CD8^+^ T-lymphocyte population. Thus, the risk of tumor escape from the immune response was reduced [34].

In our study, we showed that reprogramming combined with preliminary training increased CCR7 expression by CD8^+^ T-lymphocytes (Figure 2). CCR7 is known to be characteristic of T memory stem cells [34,35]. Interestingly, CD8^+^ T-lymphocytes showed high expression of CCR7 even in exhaustion. The stability of new properties of reprogrammed cells is important for increasing the effectiveness in cell therapy, since transplanted T-lymphocytes, including modified ones, are affected by various damaging (exhausting) factors (tumor microenvironment, cytokines) [32].

Based on the results of our in vitro experiments, we conclude that reprogrammed T-lymphocytes in LLC cell culture were more resistant to apoptosis in comparison with naive T-lymphocytes (Figure 3). On the other hand, inhibitors of PD-1 and MEK in combination with the training of T-lymphocytes allowed for the increase in cytotoxic activity of T-lymphocytes (Figure 4).

During cell therapy, transplanted T-lymphocytes should migrate to the tumor for effective implementation of cytotoxic function. We observed an increase in CD8^+^ T-lymphocyte numbers in the lungs of recipient mice with LLC after intravenous administration of CFSE-labeled reprogrammed CD8^+^ T-lymphocytes of donor mice. As shown on Figure 5, CD8^+^ T-lymphocyte numbers were more than 8.9 times higher than in cell therapy with naive T-lymphocytes.

The antitumor effect of reprogrammed T-lymphocytes was confirmed by results of our in vivo experiments. In the LLC orthotopic model, well-vascularized large tumor nodes consisted of atypical cells were found in the lungs of mice (day 7). These cells were characterized by cellular and nuclear polymorphism. Multinucleated giant cells were found in the general population of atypical cells. Many cells were in mitosis. Multiple small foci of necrosis were observed in the tumor tissue (Figure 6). Intravenous administration of reprogrammed CD8^+^ T-lymphocytes reduced the number of tumor emboli in vessels, perivascular, and peribronchial metastases in the lungs of recipient mice in LLC orthotopic model. The number of lung metastases after the treatment with reprogrammed T-lymphocytes decreased (Table 1). In the present study, we aimed to model lung cancer, since in our opinion, cell therapy with reprogrammed CD8+ T-lymphocytes can be more effective to CSCs, while radiation therapy or chemotherapy are not effective on CSCs.

We hypothesized that CSCs, as well as tumor cells, are potential targets for reprogrammed CD8^+^ T-lymphocytes. Indeed, all studied populations of CSCs (Axl^+^, Axl^+^CD117^+^, EGF^+^CD44^+^Sox2^+^, EGF^+^Sox2^+^, CD44^+^Sox2^+^, CD117^+^Sox2^+^, CD117^+^EGF^+^CD44^+^Sox2^+^) in the lungs of recipient mice significantly decreased after cell therapy (day 7) (Figure 7). The number of CSCs in the blood had declining trends. It is shown that these cells can thus act as potential contributors to tumor progression and formation of metastases. Since these cell types are present in the blood, we hypothesize that they are potential markers of the disease prognosis and targets of the treatment. 

Thus, results of our study indicate the effectiveness of reprogramming to enhance the antitumor activity of T-lymphocytes. However, we paid attention to an increase in the content of T-lymphocytes with a high proliferative potential (CD8^+^CD62L^+^CD197^+^CD45RA^+^) and memory cells (CD8^+^CD62L^−^CD44^+^) in the lungs and blood in addition to the data indicating the antitumor activity of reprogrammed CD8^+^ T-lymphocytes in vitro and in vivo (Figure 8). These data indirectly confirm CD8^+^ T-lymphocytes high proliferative activity which is important for cell therapy of lung cancer.

Our work is limited by the capacity of animal models to mimic the human lung cancer. Thus, further evaluation in more mouse experimental systems and randomized trials is needed.

## 5. Conclusions

In our study, we demonstrate that reprogramming by inhibiting the MAPK/ERK pathway through MEK1/2i and blockade of the PD/PDL-1 signaling pathway by a human monoclonal antibody, nivolumab, enhances antitumor activity of CD8^+^ T-lymphocytes in the LLC orthotopic model. At the same time, various populations of cancer stem cells are potential targets for reprogrammed CD8^+^ T-lymphocytes.

## Figures and Tables

**Figure 1 biomedicines-10-01450-f001:**
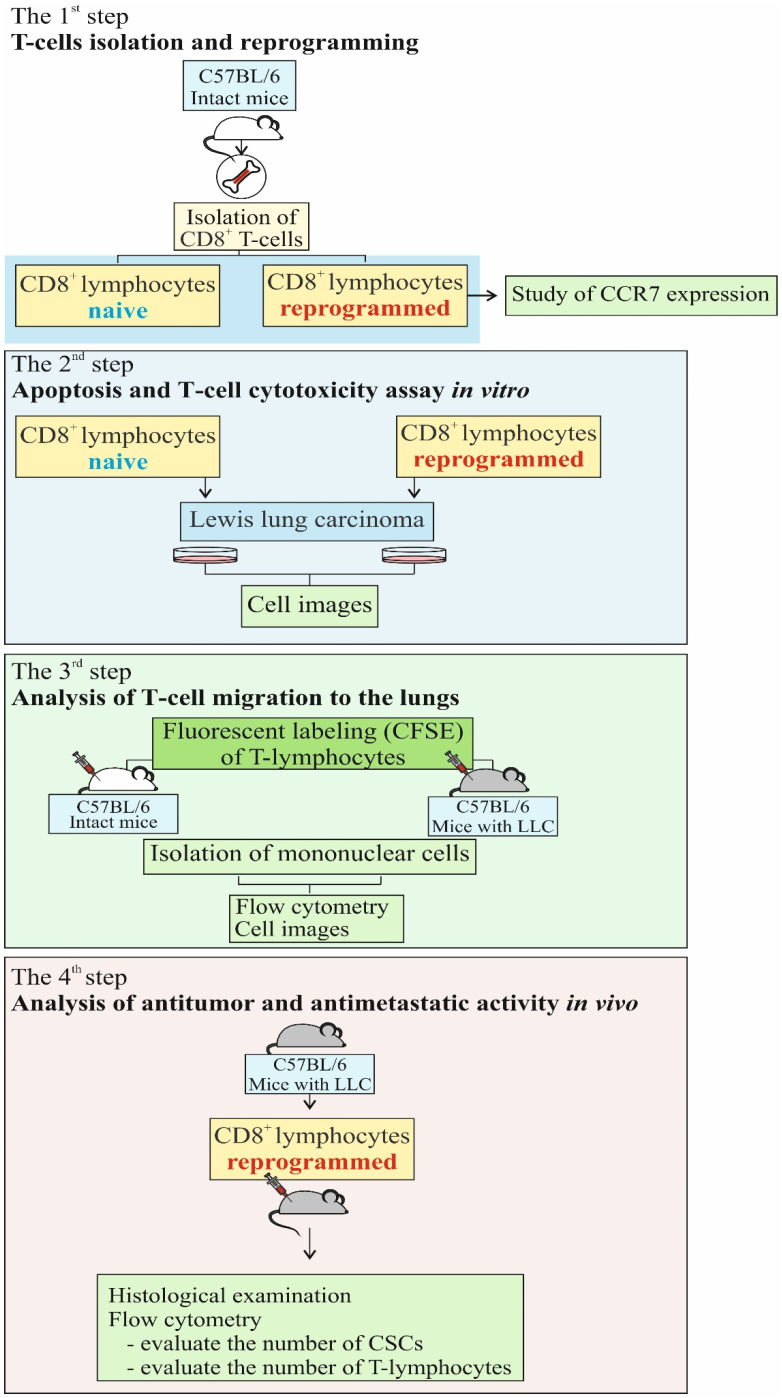
The experimental design of investigation.

**Figure 2 biomedicines-10-01450-f002:**
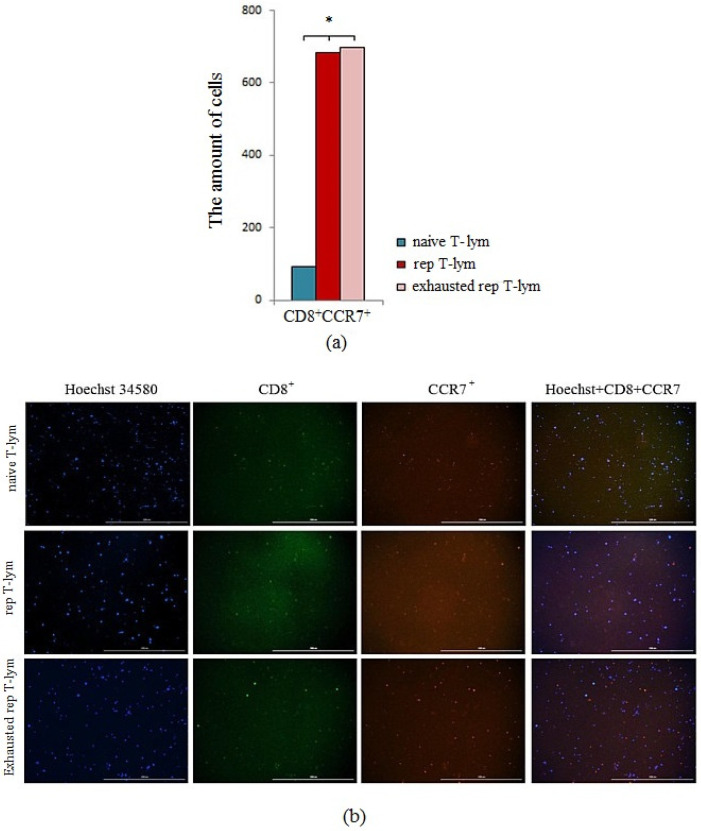
The count of CCR7^+^ T-cells in a culture of naive, reprogrammed, exhausted reprogrammed CD8^+^ T-lymphocytes isolated from the bone marrow. (**a**) The count of naive, reprogrammed, and exhausted reprogrammed CD8^+^ T-lymphocytes of the bone marrow of C57BL/6 mice expressing the CCR7 marker in T-lymphocyte culture; (**b**) 20× images of T-cells stained with: Hoechst (blue) to identify cell nuclei; CD8 FITC (green); CCR7 AF555 (red); (Hoechst^+^CD8^+^ CCR7) composite image using all three colors. Determination of the percentage of cells CD8^+^CCR7^+^ is made by the ratio of cells counted in green and red channel to total cells counted using blue (DAPI) channel. All scale bars are 1000 µm. *—for comparison with the naive T-lymphocyte by Mann–Whitney test (*p* < 0.05).

**Figure 3 biomedicines-10-01450-f003:**
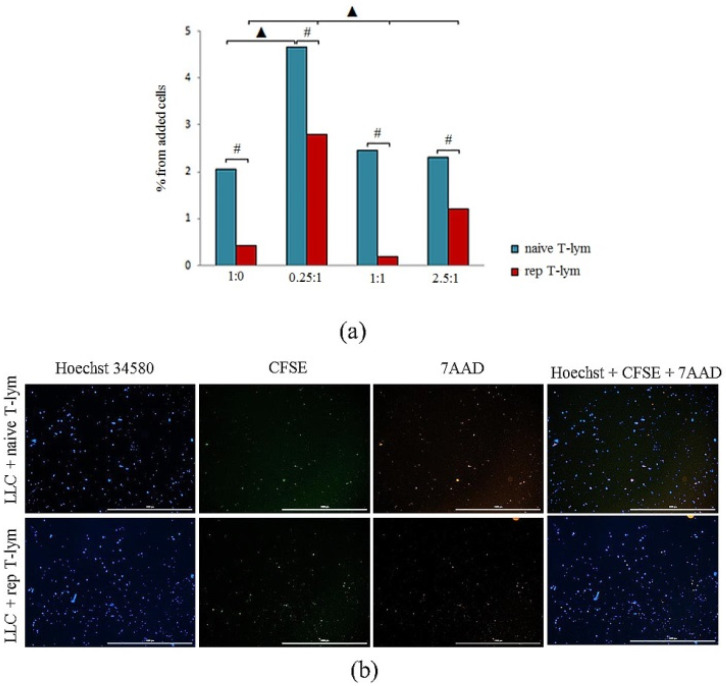
Survival of naive and reprogrammed bone marrow CD8^+^ T-lymphocytes in LLC culture (ratio of 1:1). (**a**) The count of apoptotic naive and reprogrammed CD8^+^ T-lymphocytes isolated from the bone marrow of C57BL/6 mice after coculturing with LLC (% from added cells) in ratio between T-lymphocytes and LLC 1:0, 0.25:1, 1:1 and 1.25:1; (**b**) 4× images of T-cells stained with: Hoechst (blue) to identify cell nuclei; CD8^+^ CFSE (green); 7AAD (red); (Hoechst^+^CFSE^+^7AAD) composite image using all three colors. Determination of the percentage of dead T-lymphocytes Hoechst^+^CFSE^+^ is made by the ratio of cells counted in blue and green channel to total cells. All scale bars are 1000 µm. #—for comparison with the T-lymphocyte control by Mann–Whitney test (*p* < 0.05); ▲—for comparison with the group “naive CD8^+^ T-lymphocytes + tumor cells” by Mann–Whitney test (*p* < 0.05).

**Figure 4 biomedicines-10-01450-f004:**
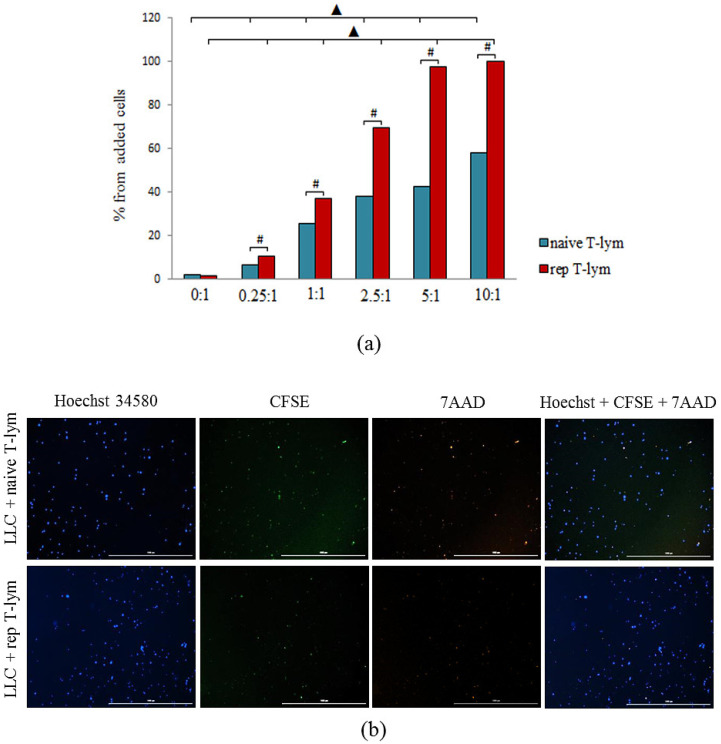
Naive and reprogrammed CD8^+^ T-lymphocytes cytotoxicity of bone marrow in LLC culture (ratio of 1:1). (**a**) The count of apoptotic tumor LLC cells after cocultivation with bone marrow CD8^+^ T-lymphocytes (% from added cells); (**b**) Hoechst (blue) to identify cell nuclei; CD8^+^ CFSE (green); 7AAD (red); (Hoechst^+^CFSE^+^7AAD) composite image using all three colors. Determination of the percentage of dead cells of LLC Hoechst+7AAD+ is made by the ratio of cells counted in blue and red channel to total cells of LLC. All scale bars are 1000 µm. #—for comparison with LLC control by Mann–Whitney test (*p* < 0.05); ▲—for comparison with the group “naive CD8^+^ T-lymphocytes + tumor cells” by Mann–Whitney test (*p* < 0.05).

**Figure 5 biomedicines-10-01450-f005:**
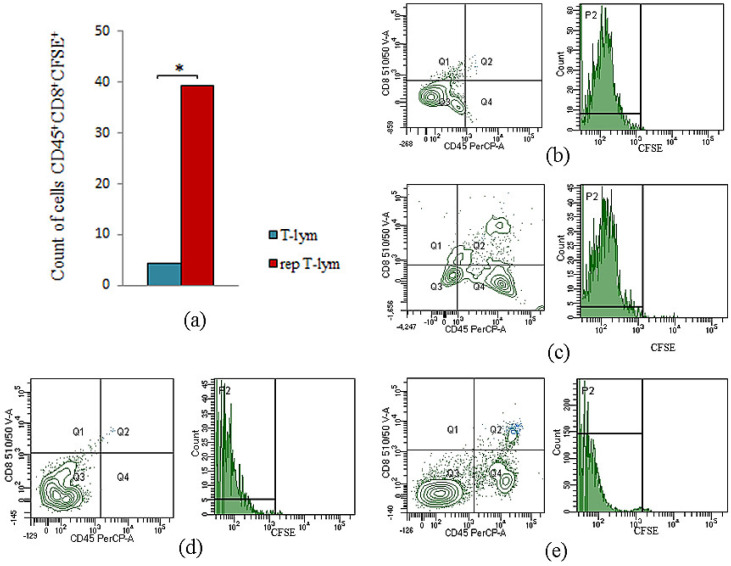
The count of CD45^+^CD8^+^CFSE^+^ cells isolated from bone marrow in the lungs of mice with LLC 60 min after administration. (**a**) The count of naive and reprogrammed CD45^+^CD8^+^CFSE^+^ cells in the lungs of mice with LLC (% from number of CD45^+^CD8^+^ T-lymphocytes); (**b**) naive CD8^+^ T-lymphocytes, unstained control; (**c**) naive CD8^+^ T-lymphocytes, stained control; (**d**) reprogrammed CD8^+^ T-lymphocytes, unstained control; (**e**) reprogrammed CD8^+^ T-lymphocytes, stained control. *—for comparison with the naive T-lymphocytes by Mann–Whitney test (*p* < 0.05).

**Figure 6 biomedicines-10-01450-f006:**
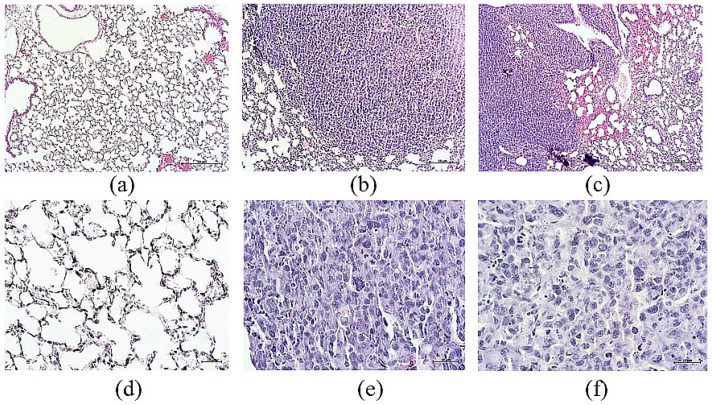
Micrographs of lung sections obtained from male C57BL/6 (**a**,**d**) mice of intact control; (**b**,**e**) mice with LLC; (**c**,**f**) mice with LLC treated with reprogrammed CD8^+^ T-lymphocytes on d7. Tissues were stained with hematoxylin-eosin. ×100 (**a**–**c**) and ×400 (**d**–**f**). Scale bar 50 μm (**a**–**c**). Scale bar 10 μm (**d**–**f**).

**Figure 7 biomedicines-10-01450-f007:**
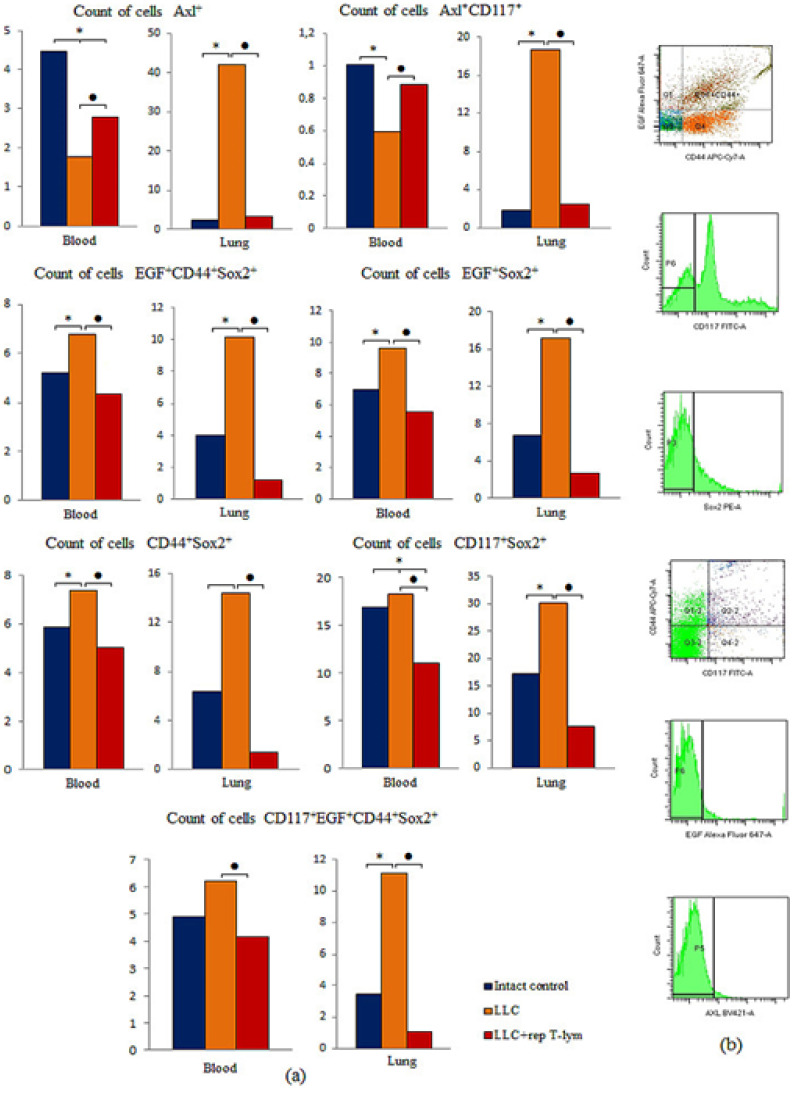
The effect of cell-therapy CD8^+^ T-lymphocytes on cancer cells and cancer stem cells. (**a**) The count (% of total mononuclear cells number) of cancer cells and cancer stem cells in blood and lung mice with LLC on d7; (**b**) phenotype establishment and qualitative analysis of CD117 (FITC), CD44 (APC-Cy7), Axl (BV421), EGF (AF647), Sox2 (PE). *—for comparison with the intact group by Mann–Whitney test (*p* < 0.05); ●—for comparison with the mice with LLC by Mann–Whitney test (*p* < 0.05).

**Figure 8 biomedicines-10-01450-f008:**
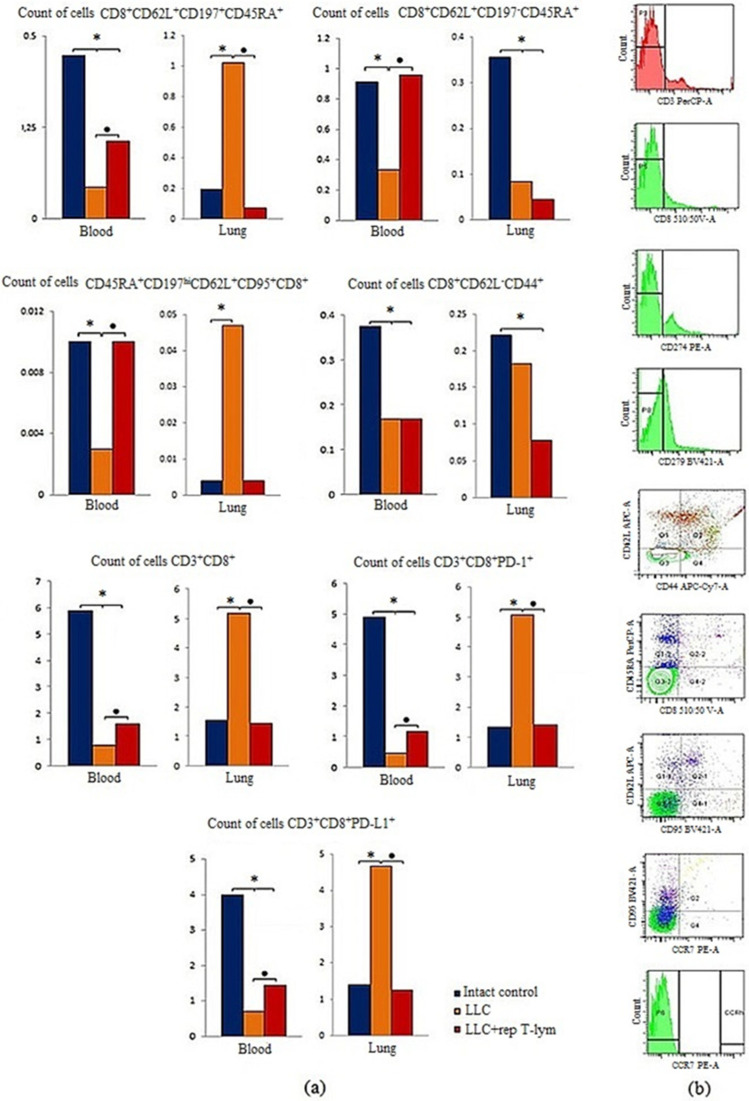
The effect of cells with reprogrammed CD8^+^ T-lymphocytes on CD8^+^ T-lymphocyte count. (**a**) The count of CD8^+^ T-lymphocytes (% of total mononuclear cells number) in the blood and lung mice with LLC on d7; (**b**) phenotype establishment and qualitative analysis of CD3 (PerCP), CD8 (BV510), CD62L (APC), CD197 (PE), CD45RA (PerCP-Cy™5.5), CD95 (BV421), PD-1(BV421), PD-L1(PE). *—for comparison with the intact group by Mann–Whitney test (*p* < 0.05); ●—for comparison with the mice with LLC by Mann–Whitney test (*p* < 0.05).

**Table 1 biomedicines-10-01450-t001:** The effect of cell therapy with reprogrammed CD8^+^ T-lymphocytes on tumor growth of mice with LLC on d7 (M ± m).

Values/Parameters	Intact Control	LLC	Reprogrammed CD8^+^ T-Lymphocytes of Bone Marrow
Tumor volume, mm^3^	0	4.62 ± 3.85 ^#^	1.95 ± 2.33 ^#,^*
The average number of metastases	0	2.50 ± 0.25 ^#^	0.50 ± 0.34 ^#,^*

Differences are significant in comparison: #—with intact control by Mann–Whitney test (*p* < 0.05); *—with pathological control by Mann–Whitney test (*p* < 0.05).

## Data Availability

Not applicable.

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
