# Peer review of "Reprogrammed CD8+ T-Lymphocytes Isolated from Bone Marrow Have Anticancer Potential in Lung Cancer"

_biomedicines, 2022, doi:10.3390/biomedicines10061450_

Round 1
Reviewer 1 Report
The paper is focused on the effects of "metabolic" reprogramming of T-lymphocytes on their anti-tumor activity. The Authors show that it prompts the maintenance of apoptosis-resistant and cytotoxic CCR7+ lymphocytes that interfere with LLC progression in vitro and in vivo. This is an interesting paper; however there are several crucial points that need consideration before the story can be accepted for publication:
- Introduction is difficult to follow. There are many types/ways of metabolic reprogramming. The Authors should describe in the 1st paragraph (before the introduction of MAPK inhibitors) what they mean by "metabolic reprogramming" in the context of the paper and why it is crucial for cancer development/therapy;
- Please, describe PD1/PD-1L system in a slightly more detailed fashion;
- The last 2 paragraphs of Intro could easily be combined, along with the more comprehensive description of study aims;
- Result section: short notes providing rational for each set of experiments and the conclusions would help the readers to follow the story;
-How was the apoptosis quantified? Method is not described;
-please, rearrange the Result section according to the scheme in Fig. 1. Now, the data from the 2nd step are described at the very end, etc.
-the Authors claim that they have metabolically reprogrammed T lymphocytes; however the do not provide any data on the metabolic profile of the cells before/after the treatment. This point should be experimentally addressed;
- Discussion is again difficult to follow. Milestones of the paper need to be pointed out and discussed point by point. For instance, the section on CSC is rather superficial, even though this point seems important for the impact of the story;
Author Response
We thank the reviewer for their time and valuable comments. We have now revised the manuscript according to the suggestions. Please find below the reviewers’ comments and our responses. All changes have been included in the revised manuscript.
The paper is focused on the effects of "metabolic" reprogramming of T-lymphocytes on their anti-tumor activity. The Authors show that it prompts the maintenance of apoptosis-resistant and cytotoxic CCR7+ lymphocytes that interfere with LLC progression in vitro and in vivo. This is an interesting paper; however there are several crucial points that need consideration before the story can be accepted for publication:
- Introduction is difficult to follow. There are many types/ways of metabolic reprogramming. The Authors should describe in the 1st paragraph (before the introduction of MAPK inhibitors) what they mean by "metabolic reprogramming" in the context of the paper and why it is crucial for cancer development/therapy;
Response
Thank you for this question. Tumor and tumor microenvironment suppress proliferation of CD8+ T-lymphocytes and decrease their numbers. During interaction with the tumor, CD8+ T-lymphocytes have a decreased activity of metabolic pathways and hyporeactive phenotypes that cannot be restored by stimulation. These changes limit the use of effector T-lymphocytes in immunotherapy of lung cancer. This is one of the mechanisms by which tumors escape from immune surveillance. Restoration of T-lymphocyte function is important for the development of an antitumor response. «Reprogramming» means directed differentiation of naive T-lymphocytes to T-lymphocytes, which have enhanced cytotoxicity and proliferative capacity.
- Please, describe PD1/PD-1L system in a slightly more detailed fashion;
Response
Programmed death protein 1 (PD1) is transmembrane receptors. It expresses on the immune cells (activated T-lymphocytes, dendritic cells, macrophages). PD1 inhibits the activity of T-lymphocytes, controlling immunological tolerance and preventing the development of autoimmune processes [Francisco L.M., Sage P.T., Sharpe A.H. The PD-1 pathway in tolerance and autoimmunity. Immunol Rev. 2010, 236, 219-242. doi: 10.1111/j.1600-065X.2010.00923.x]. PDL1 is a co-inhibitory molecule. It's highly expressed by tumor cells. PD1/PDL1 signaling leads to dysfunction of tumor-infiltrating T-lymphocytes (decreased production of cytokines, reduced cytotoxic activity and reduced survival of T-lymphocytes). The PD1/PD-1L system is one of tumor immune escape mechanisms. Thus, the effect on the PD1/PD-1L system is important for overcoming tumor resistance [Liu C., Seeram N.P., Ma H. Small molecule inhibitors against PD-1/PD-L1 immune checkpoints and current methodologies for their development: a review. Cancer Cell Int. 2021, 21, 239. doi: 10.1186/s12935-021-01946-4]
- The last 2 paragraphs of Intro could easily be combined, along with the more comprehensive description of study aims;
Response
Thank you for this comment. We combined the last 2 paragraphs and added more information about cancer stem cells.
- Result section: short notes providing rational for each set of experiments and the conclusions would help the readers to follow the story;
Response
We added short notes for each series of experiments.
-How was the apoptosis quantified? Method is not described;
Response
Thank you for your comment. We have added this information to "Materials and methods", paragraph 2.12.
-please, rearrange the Result section according to the scheme in Fig. 1. Now, the data from the 2nd step are described at the very end, etc.
Response
Thank you for your comment. We modified Fig. 1 to better to show the design of the experiment.
-the Authors claim that they have metabolically reprogrammed T lymphocytes; however the do not provide any data on the metabolic profile of the cells before/after the treatment. This point should be experimentally addressed;
Response
Thank you so much for the important comment. In present study T-lymphocytes reprogramming was based on the article "MEK inhibition reprograms CD8+ T lymphocytes into memory stem cells with potent antitumor effects", Verma et al. The authors used MEKi to made T-lymphocytes with increased CCR7 expression and enhances of mitochondrial function. These changes were called as "metabolic reprogramming". We demonstrated the increase of CCR7 expression on CD8+ T-lymphocytes, evaluated antitumor activity of CD8+ T-lymphocytes in the LLC orthotopic model in our study. This consistent with the data reported by Verma et al. We agree that to confirm metabolic reprogramming, it's necessary to assess the mitochondrial function of reprogrammed T-lymphocytes. We are going to evaluate the mitochondrial function in the next experiment. We changed "metabolic reprogramming" with "reprogramming" for greater clarity.
- Discussion is again difficult to follow. Milestones of the paper need to be pointed out and discussed point by point. For instance, the section on CSC is rather superficial, even though this point seems important for the impact of the story;
Response
Thank you for the comment. Indeed, cancer stem cells have an important role in the development and progression of lung cancer. We added more information about cancer stem cells in the "Introduction". In addition we added information about MEK inhibitor, about cancer stem cells in the «Discussion». We also found it necessary to indicate the limitations of present study.

Reviewer 2 Report
The manuscript entitled:" Reprogrammed СD8 2 +T-Lymphocytes Isolated from Bone Marrow Have Anticancer Potential in Lung Cancer" focused on the evaluation of Tlymphocytes as a potential biomarker against lung cancer is well written and requires minor considerations to be accepted for the publication:
- In the introduction section, the authors report the clinical background behind this application. In my opinion, the authros should implement this section with the therapies conventionally available for this patients; in particular for this hystotype.
- In the text, please, could the authors improve dpi for figure 1 and 2?
- Please, could the authors discuss hwo this approach may benefit in order histological classess of lung cancer patients?
- Could the authors analyze the direct relationship between their approach and clinical outcome? Coudl the authors explain that this event was dependent from the t- lymphocytes repogrammin or from inner molecular features of adopted cell lines?
Author Response
We thank the reviewer for their time and valuable comments. We have now revised the manuscript according to the suggestions. Please find below the reviewers’ comments and our responses. All changes have been included in the revised manuscript.
The manuscript entitled:" Reprogrammed СD8 2 +T-Lymphocytes Isolated from Bone Marrow Have Anticancer Potential in Lung Cancer" focused on the evaluation of Tlymphocytes as a potential biomarker against lung cancer is well written and requires minor considerations to be accepted for the publication:
- In the introduction section, the authors report the clinical background behind this application. In my opinion, the authros should implement this section with the therapies conventionally available for this patients; in particular for this hystotype.
Response
- Thank you for the interesting question. We now added the information in our manuscript. The Lewis lung carcinoma has been an important tumor model for metastatic and angiogenesis studies and neoadjuvant chemotherapy. LLC was separated from spontaneous epidermal cancer of the lungs of a mouse in 1951, and has since been used to set up animal models with tumor. It is a rapidly growing tumor and a very malignant type of epidermoid carcinoma. Our study was experimental. We observed histological picture of lung, which is specific for the early stage of LLC orthotopic model development. The formation of a significant mass of the tumor did not occur in that short period of time (7 days). We know that our study has limitations. Our work is limited by the capacity of animal models to mimic the human lung cancer. In the present study, we aimed to model the early stage of lung cancer since, in our opinion cell therapy with reprogrammed CD8+ T-lymphocytes can be more effective, while radiation therapy or chemotherapy are needed on the stage when a significant tumor mass has been developed.
- In the text, please, could the authors improve dpi for figure 1 and 2?
Response
We now improved dpi for figures 1 and 2.
- Please, could the authors discuss hwo this approach may benefit in order histological classess of lung cancer patients?
Response
- Thank you for the question.
We evaluated antitumor activity of reprogrammed T-lymphocytes in vitro and in vivo in the Lewis lung carcinoma model. Reprogrammed T-lymphocytes were resistant to apoptosis and characterized by high cytotoxicity against Lewis lung carcinoma (LLC) cells in vitro.
It is known the most common lung cancer type is represented by non-small cell lung cancer (NSCLC, corresponding to about 90% of cases of lung cancer, the remaining ones being small lung cancer) which comprises three histological subtypes: adenocarcinoma, squamous cell carcinoma and large cell carcinoma. Lewis lung carcinoma is a rapidly growing tumor and a very malignant type of epidermoid carcinoma.
Unfortunately most patients with lung cancer have advanced disease at the time of diagnosis (stage III/IV) and despite significant developments in the oncological management of late stage lung cancer over recent years, survival remains poor.
Our experiments are in no way exhaustive investigations, with more investigation needed to fully elucidate the interactions between tumour, environment and immune system during cancer development. Further investigation of these events and parallel development of preclinical models of early disease in which to study them has the potential to yield novel biomarkers for early detection of lung cancer. However, this remains challenging and is under research.
- Could the authors analyze the direct relationship between their approach and clinical outcome? Coudl the authors explain that this event was dependent from the t- lymphocytes repogrammin or from inner molecular features of adopted cell lines?
Response
- Thank you for your question. Our experimental study was shown that cancer stem cells may be target for reprogrammed CD8+ T-lymphocytes. Although cancer stem cells only constitute a low percentage of the total tumor mass, these cells can regrow the tumor mass on their own. It was shown, a high number of cancer stem cells measured at any time during treatment has been associated, in several studies, with a shorter time to progression in metastatic breast cancer, whilst a decreasing cancer stem cells number during treatment indicated therapeutic success [1]. The prognostic value of cancer stem cells in early breast cancer has then been proven. Similar findings were even reported in lung cancer [2-7].
Moreover we propose that reprogrammed CD8+ T-lymphocytes can produce highly activated and less exhausted effector CD8+ T-lymphocytes. It was shown that MEK inhibition induces TSCM cells that tend to have lower expression of activation markers on CD8+ T-lymphocytes. Furthermore, these MEKi-treated cells possess significantly improved recall response, which explains their robust effector functions [8]. We propose that using reprogrammed CD8+ T-lymphocytes can overcome the limitations of current adoptive T cell therapies, including inefficient T cell engraftment, persistence and ability to mediate prolonged immune attack.
- Nicolini, A.; Ferrari, P.; Duffy, M.J. Prognostic and predictive biomarkers in breast cancer: Past, present and future. Semin. Cancer Biol. 2018, 52, 56–73. doi: 10.1016/j.semcancer.2017.08.010
- Rivera C, Rivera S, Loriot Y, Vozenin MC, Deutsch E. Lung cancer stem cell: new insights on experimental models and preclinical data. J Oncol. 2011; 2011: 549181.
- Sarvi S, Mackinnon AC, Avlonitis N, Bradley M, Rintoul RC, Rassl DM, et al. CD133+ cancer stem-like cells in small cell lung cancer are highly tumorigenic and chemoresistant but sensitive to a novel neuropeptide antagonist. Cancer Res. 2014; 74: 1554-65.
- Alamgeer M, Ganju V, Szczepny A, Russell PA, Prodanovic Z, Kumar B, et al. The prognostic significance of aldehyde dehydrogenase 1A1 (ALDH1A1) and CD133 expression in early stage non-small cell lung cancer. Thorax. 2013; 68: 1095-104.
- Liou GY. CD133 as a regulator of cancer metastasis through the cancer stem cells. Int J Biochem Cell Biol. 2019; 106: 1-7.
- Eramo A, Lotti F, Sette G, Pilozzi E, Biffoni M, Di Virgilio A, et al. Identification and expansion of the tumorigenic lung cancer stem cell population. Cell Death Differ. 2008; 15: 504-14.
- Tirino V, Camerlingo R, Franco R, Malanga D, La Rocca A, Viglietto G, et al. The role of CD133 in the identification and characterisation of tumour-initiating cells in non-small-cell lung cancer. Eur J Cardiothorac Surg. 2009; 36: 446-53.
- Gattinoni, L. et al. A human memory T cell subset with stem cell–like properties. Nat. Med. 17, 1290–1297 (2011).

Round 2
Reviewer 1 Report
No more comments